# Lights, Camera, Interaction: Studying Protein–Protein Interactions of the ER Protein Translocase in Living Cells

**DOI:** 10.3390/ijms221910358

**Published:** 2021-09-26

**Authors:** Mark Sicking, Martin Jung, Sven Lang

**Affiliations:** Department of Medical Biochemistry and Molecular Biology, Saarland University, 66421 Homburg, Germany; mark.sicking@uni-saarland.de (M.S.); martin.jung@uks.eu (M.J.)

**Keywords:** bimolecular luminescence complementation, competition, split luciferase, membrane proteins, protein–protein interactions, Sec61 complex, Sec63, synthetic peptide complementation, TRAP complex, ER protein translocase

## Abstract

Various landmark studies have revealed structures and functions of the Sec61/SecY complex in all domains of live demonstrating the conserved nature of this ancestral protein translocase. While the bacterial homolog of the Sec61 complex resides in the plasma membrane, the eukaryotic counterpart manages the transfer of precursor proteins into or across the membrane of the endoplasmic reticulum (ER). Sec61 complexes are accompanied by a set of dynamically recruited auxiliary proteins assisting the transport of certain precursor polypeptides. TRAP and Sec62/Sec63 are two auxiliary protein complexes in mammalian cells that have been characterized by structural and biochemical methods. Using these ER membrane protein complexes for our proof-of-concept study, we aimed to detect interactions of membrane proteins in living mammalian cells under physiological conditions. Bimolecular luminescence complementation and competition was used to demonstrate multiple protein–protein interactions of different topological layouts. In addition to the interaction of the soluble catalytic and regulatory subunits of the cytosolic protein kinase A, we detected interactions of ER membrane proteins that either belong to the same multimeric protein complex (intra-complex interactions: Sec61α–Sec61β, TRAPα–TRAPβ) or protein complexes in juxtaposition (inter-complex interactions: Sec61α–TRAPα, Sec61α–Sec63, and Sec61β–Sec63). In the process, we established further control elements like synthetic peptide complementation for expression profiling of fusion constructs and protease-mediated reporter degradation demonstrating the cytosolic localization of a reporter complementation. Ease of use and flexibility of the approach presented here will spur further research regarding the dynamics of protein–protein interactions in response to changing cellular conditions in living cells.

## 1. Introduction

As the fundamental unit of life, cells endow biological systems with tremendous powers and astonishing features. In the case of differentiated eukaryotic cells, these are often compartmentalized and different membrane-surrounded or membrane-less organelles help to shape cellular fitness, metabolism, and signaling. One of the organelles that supports both intracellular signaling, for example calcium (Ca^2+^) signaling or the unfolded protein response, and secretion of proteohormones is the endoplasmic reticulum (ER) [1,2,3,4]. A major membrane protein at the crossroad of ER signaling and protein transport is the heterotrimeric Sec61 complex acting as the pore-forming component of the ER protein translocase [5,6,7].

Many different membrane-spanning (~5000) and soluble (~3000) proteins of the human proteome can be guided by the Sec61 complex to enter the secretory pathway [8]. To meet the demands for the transport of such topologically diverse precursor proteins crosslinking and native gel electrophoresis studies have demonstrated the dynamic association of the Sec61 complex with accessory factors to form the active holo-translocon [9,10,11,12]. For example, structural and proteomic data as well as pulling force studies verified the hetero-tetrameric translocon-associated protein (TRAP) complex as an important player supporting the Sec61 complex during co-translational transport of precursor proteins with an above-average glycine-plus-proline content in the signal peptide [13,14,15]. Similarly, the proteins Sec62/Sec63 were identified to support post-translational as well as substrate-specific co-translational opening of the Sec61 complex [9,16,17,18,19,20,21]. In analogy to enzyme-catalyzed reactions, auxiliary protein complexes like TRAP or Sec62/Sec63 act as allosteric effectors supporting the gating of the Sec61 complex [22].

As exemplified above for the Sec61 complex, protein translocation machineries of other organelles like the PIM complex of peroxisomes [23], the TIM/TOM complexes of mitochondria [24], or the TIC/TOC equivalents in chloroplasts [25] all represent large, heteromultimeric assemblies and their activity depends on protein–protein interactions (PPI) [26,27]. The necessity of protein multimerization for proper functionality is by no means restricted to protein translocation machineries and governs pivotal cellular processes like DNA replication and transcription, mRNA translation, transmembrane signaling mechanisms or enzymatic catalysis, and ATP production with estimates of up to 80% of proteins operating in complexes [28,29]. Typically, PPI control the assembly of protein complexes via non-covalent contacts between the side chains of amino acids from different polypeptide strands and can form large-scale networks, often referred to as the interactome [30,31]. Many methods have been developed to identify interactions of proteins and define the interactome of cells. With regard to the experimental setup, those methods can be classified as in vitro, in vivo, or in silico and include popular examples like coimmunoprecipitation, the yeast-two-hybrid system, and structure-based prediction via algorithms, respectively. However, method-specific strengths and weaknesses depending on the scientific question asked is a feature they all share [29,32,33,34]. One of the more challenging aspects when studying PPI is their dynamic nature that can trigger the transient or stable formation of homo- or heterooligomers whose association can be influenced by various cellular cues. A textbook example of a dynamic PPI of a heterooligomer regulated by cellular cues is the protein kinase A (PKA). PKA represents a soluble tetramer of two catalytic subunits harboring kinase function and two regulatory subunits that act as cAMP sensors. Separation of the catalytic from the regulatory subunits activates the kinase function and is influenced by intracellular (cAMP) and extracellular (hormones, e.g., glucagon) cues, which together with the corresponding hormone receptor and adenylate cyclase represent critical elements of a signal transduction pathway [35]. Bimolecular protein-fragment complementation assays are well suited to detect the spatiotemporal dynamics of binary PPI like that of PKA in intact living cells. These assays are based on proteinaceous split reporters with the complementary reporter fragments genetically fused to proteins of interest, whose interaction can be directly visualized [36].

Here we used bimolecular luminescence complementation (BiLC) as live cell assay to further complement the studies that so elegantly demonstrate the association of the Sec61 complex with the TRAP complex or Sec62/Sec63 either in silico, in vitro, or after reconstitution or vitrification. In the case of BiLC, the interaction of tagged protein pairs reassembles a functional, split luciferase and can be visualized upon addition of a luciferase substrate [37,38]. After verifying the dynamic PPI of the catalytic and regulatory subunits of PKA, our proof-of-concept shows “intra-complex” interactions between subunits of the Sec61 complex (Sec61α–Sec61β) as well as of the TRAP complex (TRAPα–TRAPβ). Furthermore, we demonstrate “inter-complex” interactions between membrane proteins of different complexes such as Sec61α–TRAPα, Sec61α–Sec63, and Sec61β–Sec63. Both types of PPI, intra- and inter-complex ones, can be perturbed by competitive over-expression of untagged variants. Alternatively, interactions of membrane or soluble protein pairs that rely on reassembly of the split luciferase components in the cytosol are abolished by the combination of plasma membrane permeabilization and trypsin-mediated reporter degradation.

## 2. Results

To reliably verify the PPI of ER membrane proteins in living cells, we established a microplate reader–based bimolecular protein-fragment complementation assay widely used for investigations of protein interactions [33]. The split reporter used here is an optimized variant of the catalytic subunit of the luciferase derived from deep sea luminous shrimp called NanoLuc [38]. The 19 kDa polypeptide consisting of a ten-stranded β-barrel topology is separated after the ninth β-barrel providing an 18 kDa and 1 kDa fragment called LgBiT and SmBiT, respectively. Although the two fragments can reconstruct a functional luciferase that provides a bright luminescence in the presence of its substrate furimazine, the engineered LgBiT and SmBiT show a low intrinsic affinity for each other. The association constant of LgBiT and SmBiT has a k_D_ of 190 μM, which is above the k_D_ of most physiological relevant PPI [37]. Thus, when LgBiT and SmBiT are added as fusion tags to proteins of interest, luminescence will occur if two prerequisites are met: (i) the presence of furimazine and (ii) direct interaction, i.e., close proximity of the proteins of interest (Figure 1A).

### 2.1. Establishing the NanoBiT Assay Based on the cAMP-Dependent Protein Kinase A, Forskolin, and Semi-Permeabilization

The interacting regulatory and catalytic subunits of protein kinase A (rPKA, cPKA) were C-terminally tagged with the LgBiT (_L_) and SmBiT (_S_) fragments, respectively. Corresponding fusion constructs were called rPKA-C_L_ and cPKA-C_S_. The non-interacting HaloTag-based fusion construct Halo-C_S_ served as negative control (Figure 1A). HeLa cells were seeded in a 96-well format and left untreated or were transfected for 24 h. Besides mock transfection, cells were transfected with the plasmid encoding for rPKA-C_L_ alone or in combination with either Halo-C_S_ or cPKA-C_S_. Four minutes after addition of furimazine, the luminescence units (LU) of the five conditions were measured. As expected, strong luminescence was detected only for the bona fide interacting fusion proteins rPKA-C_L_ and cPKA-C_S_ providing a more than 600-fold brighter luminescence than the rPKA-C_L_ and Halo-C_S_ pair (Figure 1B). The reversibility of the rPKA and cPKA interaction was demonstrated using forskolin, a non-selective activator of most adenylate cyclase isoforms [39]. Conversion of ATP to cAMP by forskolin-activated adenylate cyclases caused the rapid disassembly of rPKA from cPKA. Accordingly, forskolin treatment 10 or 20 min after addition of the NanoLuc substrate furimazine induced a rapid decrease of luminescence (Figure 1C). As depicted by the vertical gray bars in Figure 1C the addition of furimazine and all other substances required their manual application and caused a short measurement gap that was kept constant using a time window of one minute. The same applies to other measurements shown later. We also tweaked the system for the use of larger, membrane-impermeable effectors including soluble enzymes such as trypsin. Cells expressing the rPKA-C_L_ and cPKA-C_S_ reporter pair as representation of a verified PPI were first subjected to permeabilization of the plasma membrane by digitonin and subsequently treated with trypsin for proteolytic cleavage of the reporter proteins to eliminate luminescence. As shown in the line graphs in Figure 1D, addition of the substrate caused a sharp increase in luminescence within 4 to 5 min (treatment 1). Somewhat surprisingly, the application of digitonin for permeabilization of the plasma membrane (treatment 2, red or blue line) further increased signal intensity. We attributed this effect to improved access of the bulky furimazine substrate to the re-assembled luciferase. The subsequent addition of the protease trypsin after 40 min (treatment 3, red line) or 70 min (treatment 4, blue or black line) eliminated luminescence almost entirely due to protease-mediated digestion of the fusion proteins (Figure 1D). Of note, trypsin treatment of intact cells prior to permeabilization did not cause a significant drop in luminescence due to reporter degradation or eventual detachment of adherent cells from the microplate surface (cf. Figures 5B and 6B, blue lines). Correspondingly, using a non-proteolytic enzyme like RNase A further corroborated the specific elimination of luminescence by trypsin-mediated reporter degradation (Appendix A).

This line of experiments showed the applicability and dynamics of the BiLC system for cytosolic interaction partners in living cells. Permeabilization of the plasma membrane by digitonin also granted access of otherwise membrane-impermeable agents including large biological effectors such as trypsin.

### 2.2. Expression Profiling via Synthetic Complementation Using Chemically Synthesized Low and High Affinity Oligopeptides

An often-ignored issue with protein-fragment complementation assays is the sufficient expression of fusion constructs and synthesis of the encoded fusion proteins, which is equally relevant for constructs encoding soluble and membrane proteins. Therefore, before applying BiLC to ER membrane proteins, we deconstructed the system and designed an assay for measuring expression of the LgBiT fusion constructs, when the LgBit is located in the cytosol (Figure 2A). Instead of expressing the 11 amino acids (aa) long SmBiT peptide as a fusion construct in parallel with a LgBiT fusion protein, we chemically synthesized the SmBiT peptide and dissolved it in DMSO (Appendix A). During synthesis, the SmBiT peptide was N-terminally extended by a single cysteine for further downstream processes (see below). Like the 11 + 1 aa SmBiT peptide, we also synthesized a scrambled version thereof and an independent actin peptide (16 aa) as two negative controls that should not reconstitute a functional luciferase in presence of a genetically expressed LgBiT fusion protein. Using the semi-permeabilization protocol with digitonin (cf. Figure 1D), cells expressing the rPKA-C_L_ construct were provided 100 µM of a negative control peptide (scrambled or actin) or the SmBiT. As expected, only upon the addition of the SmBiT peptide did a complementation with rPKA-C_L_ occur (Figure 2B). Three points are worth emphasizing. One, taking into consideration the inefficient association between LgBiT and SmBiT (k_D_ = 190 μM) the use of a high peptide concentration like 100 µM was anticipated. Two, titrating the concentrations of the SmBiT peptide (1, 10, 100 µM) showed a clear dose-dependence of complementation and luminescence. Three, even at a concentration of 100 µM, the synthetic SmBiT provided a 10-fold lower signal intensity than the genetically expressed PPI pair rPKA-C_L_ and cPKA-C_S_ (Figure 2B, dark blue versus gray line).

During the initial mutational screens for low affinity versions of the SmBiT, Dixon and colleagues also reported variants with exceptionally high affinities for the LgBiT in the nanomolar range [37]. For one of the variants the glutamates in position 8 and 9 of the original SmBiT undecapeptide were exchanged by lysines. We also synthesized this peptide and based on its high affinity called it HaBiT (Appendix A). As shown in Figure 2C, after semi-permeabilization, HaBiT was also suited for expression profiling of the rPKA-C_L_ construct in a dose-dependent manner. Due to the high affinity, HaBiT interacted with rPKA-C_L_ in a concentration range from 10 to 100 nM providing efficient luminescence (Figure 2C and Appendix A). At 100 nM the HaBiT signal exceeded that of the genetically encoded reporter pair rPKA-C_L_ and cPKA-C_S_. Combining the data of dose-dependence and tested concentrations for the synthetic SmBiT and HaBiT peptide, the latter showed an approximately 10,000-fold higher activity in the expression profiling assay (Figure 2D). Taken together, both peptides SmBiT and HaBiT can be used to verify and compare steady-state expression of generated fusion constructs.

### 2.3. Verifying Expression of LgBit Fusion Constructs of ER Protein Translocase Subunits via Synthetic Peptide Complementation

Before testing the PPI of ER membrane proteins in living cells, the synthetic complementation approach was used to verify synthesis of the LgBiT fusion proteins 24 h post transfection. In addition to the cytosolic positive control rPKA-C_L_, five N- or C-terminally LgBiT-tagged ER membrane proteins were tested including Sec61α, Sec61β, TRAPα, and TRAPβ (Figure 3A). The latter two are classical type I membrane proteins with one N_exo_-C_cyto_ transmembrane helix (TMH) and a cleavable signal peptide [40]. Sec61β is a type IV or tail-anchored membrane protein with a single TMH at the very C-terminus inserted in an N_cyto_-C_exo_ orientation [41]. Sec61α has 10 TMH with its N- and C-terminus located in the cytosol [42]. All six of the LgBiT fusion proteins were tested with three different peptides (cf. Figure 2A). While the scrambled peptide served as negative control and for normalization, the SmBiT and HaBiT peptide served as positive controls that allowed reconstitution of a functional luciferase in case the LgBiT fusion construct was properly expressed. Taking the different affinities of SmBiT and HaBiT for the LgBiT into consideration, different peptide concentrations had to be used. Scrambled and SmBiT were added to a final concentration of 100 µM, whereas HaBiT was used at 100 nM (Figure 3B). Despite the three orders of magnitude lower concentration of HaBiT, it provided a 5–10-times stronger signal in all six tested synthetic complementations. The strongest signal was achieved in combination with the cytosolic rPKA-C_L_. Compared to the scrambled peptide, four out of five LgBiT-tagged membrane protein fusions (Sec61α-N_L_, Sec61β-C_L_, TRAPα-C_L_, TRAPβ-C_L_) provided a more than 10-fold stronger luminescence when complemented with SmBiT peptide. Only the Sec61α-C_L_ protein showed lower complementation efficiency of ~2 relative luminescent units (RLU), likely caused by its lower abundance (Figure 3B). When the HaBiT peptide was used in combination with the ER membrane protein fusions, the RLU were at least 50-fold higher in comparison to the scrambled peptide, with the exception of the Sec61α-C_L_ protein peaking at 15 RLU. Thus, the synthetic complementation approach substantiated the plasmid-driven production of the LgBiT-tagged ER membrane proteins with the Sec61α-C_L_ protein providing a low complementation likely reflecting the limited synthesis or stability of this fusion protein.

Despite its ER luminally located LgBiT tag, the Sec61β-C_L_ protein also responded strongly using the SmBiT or HaBiT peptide after semi-permeabilization (Figure 3). While an inverted topology of Sec61β-C_L_ could be one explanation, other reasons might entail unintentional permeabilization of the ER membrane by digitonin or the active transport of the synthetic oligopeptides into the ER by the transporter associated with antigen processing, TAP [43]. As soon as more sensors with luminally located LgBiT tags are available, further experiments will test the impact of TAP, permeabilization, and orientation more systematically.

### 2.4. Validating Expression of SmBit Fusion Constructs via Western Blotting Using a Polyclonal Antibody Raised against the SmBiT

The BiLC approach relies on the presence of both types of fusion proteins allowing the 1:1 stoichiometric interaction of the LgBiT and SmBiT tags. After confirming expression of the LgBiT-tagged constructs via synthetic complementation (Figure 3), we also tested for the sufficient expression of the SmBiT-tagged fusion constructs via Western blotting. Therefore, we raised a polyclonal rabbit antibody against the SmBiT undecapeptide, which required the aforementioned N-terminal cysteine for coupling of the SmBiT to the immunogenic keyhole limpet hemocyanin. As the α-SmBiT antibody showed unspecific binding and the occurrence of background bands, mock transfected cells served as negative control and helped to verify the correct signal of the individual SmBiT-tagged fusion proteins. First, the cytosolic control fusions cPKA-N_S_ and Halo-C_S_ were probed for abundance on protein level. Both proteins provided an additional band in comparison to the mock transfected cells and confirmed proper expression of the constructs (Figure 4). Likewise, the SmBiT fusion proteins TRAPα-C_S_, TRAPβ-C_S,_ Sec61α-C_S_, and Sec61β-N_S_ could be detected using the generated α-SmBiT antibody (Figure 4). We also generated two SmBiT fusions of the much larger Sec63 protein, Sec63-C_S_ and Sec63-N_S_. Neither of those two constructs could be detected with the α-SmBiT antibody. We attributed the lack of detection to the more pronounced occurrence of unspecific cross-rations of the α-SmBiT antibody in the molecular weight range above 40 kDa which might obscure efficient detection of the Sec63 fusion protein at approximately 100 kDa. Further, the use of polyclonal antibodies raised against the native portion of the Sec63 fusion proteins did not provide an additional band of higher molecular weight likely due to a combination of the small size of the SmBiT tag, the intended low expression of fusion constructs, and partial masking of the antibody-epitope by addition of the tag. The next alternative for testing expression of the SmBiT fusion construct was to test their co-expression together with a LgBiT as part of a functional PPI experiment. Luminescence arising from a combination of a LgBit and a SmBiT fusion protein would indicate synthesis of both components as shown in the following section.

### 2.5. α- and β-Subunit Interactions of the Sec61 or TRAP Complex

Next, we addressed the direct PPI of ER membrane protein pairs tagged with SmBiT and LgBiT. As proof of concept, we tested the intra-complex interaction of the heterotrimeric Sec61 complex. In its native environment, the Sec61complex consists of the channel-forming Sec61α subunit flanked by the two tail-anchored subunits β and γ [44,45]. As depicted in Figure 5A, we first tested if the cytosol facing N-termini of Sec61α and Sec61β are juxtaposed and do interact. After transfection of cells with the corresponding reporter constructs (Sec61α-N_L_, Sec61β-N_S_) and addition of the NanoLuc substrate luminescence was detected (Figure 5B). As seen before for the cytosolic PKA constructs (Figure 1C), light emission for the Sec61α-N_L_ and Sec61β-N_S_ pair was peaking 4 min past furimazine addition. To exclude drastic mislocalization of the constructs and verify that the Sec61α–Sec61β interaction occurs intracellularly, cells were subjected to trypsin-mediated reporter degradation. Indeed, only after permeabilization of the plasma membrane by digitonin, the luminescent signal was eliminated by addition of trypsin digesting the intracellular reporters. Instead, when cells were treated with DMSO prior to trypsin, no sharp decline of luminescence could be detected upon addition of the membrane impermeable protease (Figure 5B). Similar to what was observed before, the mild digitonin treatment amplified luminescence likely by providing easier access of furimazine to the cytosol (cf. Figure 1D, Figure 2B and Figure 5B). To further substantiate the validity of the PPI between Sec61α and Sec61β, a strategy based on competition was used. In addition to transfection with the reporter pair, cells received a third plasmid encoding an untagged, wild typic variant of either interaction partner. In contrast to the reporter pair plasmids carrying the HSV-TK promotor, the plasmids used for competition with untagged variants harbored a CMV promotor generally allowing stronger expression of a downstream gene [46]. An empty vector (EV) transfection was run in parallel and served as negative control. Contrary to the non-interacting Sec61α-N_L_ and Halo-C_S_ pair, a strong luminescent signal was detected for both the Sec61α-N_L_ and Sec61β-N_S_ pair as well as the Sec61α-N_L_ and Sec61β-N_S_ plus EV transfection (Figure 5C). Notably, introducing either untagged Sec61α or Sec61β into the system luminescence of the Sec61α-N_L_ and Sec61β-N_S_ reporter pair was strongly reduced and remained almost at the background level of the negative control pair Sec61α-N_L_ and Halo-C_S_ (Figure 5C). Thus, increasing the protein level of one of the interaction partners as untagged variant significantly competed with the interaction of the tagged proteins and underlines the authenticity of the tested Sec61α–Sec61β interaction in living cells at the ER (Figure 5D).

We also tested the intra-complex PPI of the heterotetrameric TRAP complex. Three of the four TRAP subunits (α, β, δ) traverse the ER membrane as single-pass type I membrane protein. Together with the tetra-spanning γ subunit these four membrane proteins tightly associate and form the native TRAP complex that supports as allosteric effector gating of the Sec61 complex [14,15,40,47]. Regarding TRAP, we focused on the interaction between the α- and β-subunit by generating the C-terminally tagged variants TRAPα-C_L_ and TRAPβ-C_S_ (Figure 6A).

Consistent with the Sec61 complex, we subjected the TRAPα/β reporter pair to the same luminescence measurements including trypsin-mediated reporter degradation and competition with the untagged counterparts (Figure 6). When cells transfected with TRAPα-C_L_ and TRAPβ-C_S_ were subjected to the furimazine–digitonin–trypsin, protocol we saw (i) the direct interaction between TRAPα and TRAPβ, (ii) the amplified luminescence after permeabilization of the plasma membrane, and (iii) rapid loss of the signal and PPI due to the proteolytic digestion (Figure 6B and Appendix A). Like the Sec61α/β reporter pair, also TRAPα-C_L_ and TRAPβ-C_S_ showed a strong interaction in comparison to the non-interacting control pair TRAPα-C_L_ and Halo-C_S_ (cf. Figure 5C and Figure 6C). This interaction was further tested using the competition setup based on plasmid-driven expression of the untagged *TRAPA* or *TRAPB* gene. Increasing the protein level of TRAPα or TRAPβ caused a significant reduction of the PPI between TRAPα-C_L_ and TRAPβ-C_S_ (Figure 6C,D). Furthermore, we tested if the expression efficiency of the LgBiT reporter construct TRAPα-C_L_ was impaired by the additional expression of the untagged counterpart TRAPα or the presence of the EV in the transfection mix. The underlying reason was that the addition of a third plasmid during the transfection procedure might reduce transfection and/or expression efficiency of the plasmids. As the exogenously added high-affinity HaBiT peptide was able to substitute for and replace the genetically encoded SmBiT fusion construct of a given reporter pair, the synthetic peptide complementation setup was used to test for expression of the TRAPα-C_L_ during transfections with two or three plasmids. As shown in Appendix A, the presence of an additional plasmid in the mix was not affecting expression of the TRAPα-C_L_ construct, which still showed efficient complementation with HaBiT even in the presence of the EV or of the plasmid encoding the untagged TRAPα protein. Compared to the negative control pair TRAPα-C_L_ and Halo-C_S_, all other three conditions combining TRAPα-C_L_ with TRAPβ-C_S_ alone or in combination with either EV or TRAPα provided the same level of luminescence upon addition of HaBiT (Appendix A).

As Figure 5 and Figure 6 showed the broad applicability of the BiLC approach for ER membrane proteins, two questions arose. First, is the assay suited to detect inter-complex interactions, for example between Sec61 and TRAP? Second, can the assay detect PPI arising from reconstitution of the split luciferase either in the cytosol or the ER lumen? Detection of luminescence emanating from the ER lumen would further prove the applicability of the system to study PPI between membrane and/or soluble proteins occurring in the ER lumen.

### 2.6. Inter-Complex Interactions between Sec61 and TRAP as well as Sec61 and Sec63

The generated set of fusion constructs was suited to address the questions of inter-complex interaction as well as detection of ER luminal emitted luminescence. Based on the vicinity of the TRAP and Sec61 complex in native membranes of different organisms and specimens, we first tested the inter-complex interaction between the α-subunits of TRAP and Sec61 [15]. As before, cells were co-transfected with the reporter pair Sec61α-N_L_ and TRAPα-C_S_ (Figure 7A). Co-transfection of cells with Sec61α-N_L_ and Halo-C_S_ served as negative control. After addition of the luciferase substrate furimazine, strong luminescence could be detected for the inter-complex interaction pair between Sec61α and TRAPα, but not for the negative control (Figure 7B). To verify the specificity of the Sec61α–TRAPα interaction in the native, cellular context additional complementation experiments were carried out. In addition to the Sec61α-N_L_ and TRAPα-C_S_ reporter pair cells were transfected with an EV or *SEC61A1* encoding plasmid. In comparison to the additional EV transfection the expression of the untagged *SEC61A1* caused a significant reduction of luminescence (Figure 7C). Thus, increasing the amount of untagged Sec61α competed with the Sec61α-N_L_ and TRAPα-C_S_ interaction. This competition by the untagged Sec61α protein substantiated the PPI of Sec61α and TRAPα in living cells.

Next, we used a set of proteins whose LgBiT and SmBiT tags were both located in the ER lumen and belong to different translocon subcomplexes, Sec61β-C_L_ and Sec63-N_S_ (Figure 7D). When cells were co-transfected with the corresponding plasmids application of furimazine indicated (i) the functionality of the ER luminally assembled luciferase converting furimazine to furimamide plus light, and (ii) the inter-complex interaction between Sec61β and Sec63 (Figure 7E). Analogous to the TRAPα–Sec61α interaction, the Sec61β–Sec63 PPI was also susceptible to competition by expression of the untagged interaction partner. When cells carrying the interaction pair Sec61β-C_L_ and Sec63-N_S_ were transfected with a plasmid encoding wild type *SEC61B*, luminescence was much reduced compared to the corresponding EV transfection (Figure 7F).

### 2.7. Sterically Impossible Complementations Set the Threshold for Authentic Protein Interactions

Excluding the positive and negative control based on the PKA reporter pair (Figure 1), 35 combinations of LgBiT- and SmBiT-tagged proteins were tested and the signal intensity of interactions is summarized as heatmap in Figure 8A. As proof of concept, we mostly focused on well-established interactions between proteins of soluble and membrane-standing multimeric complexes like PKA, TRAP, and Sec61. Other than the interaction of the cPKA and rPKA subunits, nine PPI of ER membrane proteins were identified (Figure 8A). Undoubtedly, difficult to control methodological constraints such as transfection and expression efficiency of individual cells as well as transfection uniformity across a cell population can cloud deeper interpretation of the luminescence signal intensity of a given PPI. Yet, it is tempting to consider the emitted light intensity as surrogate marker and as first approximation for the strength of a tested PPI. The three strongest, or at least most light-intense, interactions of ER membrane proteins included the intra-complex interaction between Sec61α–Sec61β (141 RLU) and TRAPα–TRAPβ (91 RLU) as well as the inter-complex interaction Sec61β–Sec63 (202 RLU). Compared to the 613 RLU measured for the soluble rPKA–cPKA pair, those RLU are 3–6 times lower and might be explained by the limited structural flexibility and/or local mobility of membrane proteins (Figure 8B). Including those three, the validity of the five strongest PPI of ER membrane proteins was further confirmed by competition experiments with untagged proteins and is highlighted in the heatmap by check marks (Figure 5, Figure 6 and Figure 7, Figure 8A and Appendix A).

As addendum to the previous data with the LgBiT and SmBiT tags of fusion proteins being located in the same compartment, the heatmap also features the sterically impossible interactions pairs that were tested (Figure 8A, hachures). By definition, physical separation of the reporter tags by a lipid bilayer should prevent their complementation. As assumed, all ten tested pairs with LgBiT and SmBiT tags placed on opposite sides of the ER membrane provided only background luminescence and, thus, served as reliable negative controls to set the threshold value for a meaningful PPI above 10 RLU (Figure 8A hachures and Figure 8B). In light of a potential degradation, mislocalization, or inverted orientation of tagged membrane proteins, luminescence below threshold can provide useful information, too. For example, the Sec61β-C_L_ reporter represents a C-terminally extended tail-anchored protein (cf. Figure 3A). Adding 176 aa (LgBiT plus linker) shifts the single TMH of Sec61β from the C-terminus to the center of the protein and might affect topology [48,49,50]. On the one hand, Sec61β-C_L_ did not show interaction with the five tested cytosolic SmBiT variants of proteins (Sec61α/β-C_S_, TRAPα/β-C_S_, Sec63-C_S_), most of which are known to associate in close proximity to Sec61β in native membranes and, thus, speaking against an inverted topology of Sec61β-C_L_ [7,19,51]. Instead, steady-state synthesis of the Sec61β-C_L_ protein (Figure 4) as well as strong interaction with the luminally located Sec63-N_S_ (Figure 7) were detected. On the other hand, of the sterically impossible interactions the Sec61β-C_L_ plus Sec61α-C_S_ reporter pair provided the highest RLU with 8.2, which might indicate a negligible fraction of Sec61β-C_L_ proteins oriented in an inverted topology (Figure 8B). Based on these data, we considered 10 RLU as a reasonable threshold to differentiate between the luminescence of an unspecific interaction (RLU < 10) and that of valid PPI of ER membrane proteins (RLU > 10).

Taken together, BiLC provided a robust and versatile method to detect topology-dependent PPI of various ER membrane proteins including associations within heteromultimeric complexes such as TRAP or Sec61 as well as inter-complex subunit connections between Sec61 and TRAP or Sec61 and Sec63/Sec62. Authenticity of the described interactions was further corroborated by competition experiments. Increasing the protein level of one of the interaction partners as untagged protein reduced luminescence, i.e., PPI.

## 3. Discussion

Using a split luciferase and the principle of BiLC, multiple PPI of ER membrane proteins were probed in living cells. The luciferase used here represents an optimized variant of the catalytic subunit of the deep sea luminous shrimp (*Oplophorus gracilirostris*) enzyme and was called NanoLuc. The original 27 aa long signal peptide was substituted by the MV dipeptide. While the wild type luciferase had one cysteine residue close to the C-terminus, mutational analysis of the SmBit tag replaced this cysteine by phenylalanine [37,38]. Thus, despite the presence of protein disulfide isomerases and the oxidative environment found in the ER, the lack of cysteine residues in the NanoLuc should allow proper assembly of the functional monooxygenase also in this compartment. Indeed, luminescence of the re-assembled luciferase could be detected in the cytosol as well as the ER lumen. The latter was demonstrated by the Sec61β–Sec63 interaction. This interaction demonstrated (i) the efficient reconstitution of the functional luciferase in the ER lumen as well as (ii) that the oxygen- and furimazine-dependent luminescence is not blocked by the presence of an additional membrane or the milieu of the mammalian ER lumen (Figure 7D–F). However, future experiments will test more PPI for which the topological layout of the reporter pair requires reconstitution of the split luciferase in the ER lumen. In addition, considering the N-terminal signal peptide of the original shrimp luciferase, at least the wild typic version of the enzyme was adapted to the secretory pathway.

Dynamic adaptability and reversibility of PPI in living cells was demonstrated for assembly of the PKA subunits using forskolin as an activator of the adenylate cyclase (Figure 1). Thus, the system will be well suited to address the dynamics of the detected translocon interactions under different cellular conditions or perturbations. For instance, Snapp and colleagues designed a FRET assay that was used to detect ribosome associations of translocon proteins via antibody accessibility before and after RNase A treatment [52,53]. Instead of trypsin-mediated reporter degradation (Figure 1D, Figure 5B, and Figure 6B), RNase A can be used upon semi-permeabilization to test either alterations in the PPI patterns of intra- and inter-complex associations or the efficiency of the synthetic complementation with the SmBiT and HaBiT peptide. Similarly, but without the necessity of semi-permeabilization, the effect of specific ribosomal or translocon inhibitors mimicking different stages of the protein transport process can be tested [54,55,56,57].

Luminescence data from the heatmap (Figure 8A) are also displayed as interaction network summarizing the intra- and inter-complex PPI of the five ER proteins tested here (Figure 8C). Even on this small scale some central nodes and connections within the holo-translocon become evident. As expected, Sec61α represents a major node with connections to Sec61β, TRAPα, and Sec63. All of those Sec61α-related connections have previously been observed by X-ray crystallography, cryo-electron microscopy/tomography, native gel electrophoresis, co-immunoprecipitation, or cross-linking studies [10,12,15,58,59,60,61,62,63]. However, none did so in living mammalian cells. Intriguingly, using different types of substrates the existence, and eventually dynamic assembly, of substrate-matched translocon sub-complexes was demonstrated [9]. As proposed earlier, the RLU measured in living cells might be indicative of the strength and permanency of a given PPI. This idea fits reasonably well with both cross-linking data and in situ structures. While the Sec61α–Sec61β (141 RLU) pair is consistently observed being associated, the Sec61α–TRAPα (62 RLU) association was seen to dissipate when substrates like prion protein or ERj3 engaged the translocon. In turn, those substrates, which probably present a minor fraction of the totality of transported cargos, rather stimulate the Sec61–Sec63 complex formation in mammalian settings [9]. Accordingly, we found Sec61α–Sec63 as weak or transiently interacting pair with only 17 RLU. Based on the recent structural data from the yeast translocon, the Sec61 complex and Sec63/Sec62 module assemble to form a translocon for the post-translational protein transport, a mode of transport less frequently encountered in mammalian cells [16,64,65]. Yet another recent structure of the fully closed yeast Sec complex from the Park lab shows juxtaposition of the Sec61β C-terminus and Sec63 N-terminus in the ER lumen [66]. While we measured only a weak association between Sec61α and Sec63, it was the reporter pair of Sec61β C-terminus and Sec63 N-terminus that provided strong luminescence of 202 RLU (Figure 8B). Maybe Sec61β acts as the mediator that keeps Sec63 in vicinity of the Sec61 complex during transport events not relying on the Sec63/Sec62 module. Once substrates with special requirements start the gating process, Sec61β allows rapid integration of Sec63 into the active translocon. If so, Sec61β and Sec63 eventually will show a partially overlapping substrate-spectrum, which remains to be seen [6,18,67]. This scenario is reminiscent of a previous report showing the recruitment of the signal peptidase complex to the active, ribosome-engaged translocon by Sec61β [68].

TRAPα emerged as a second node in the interaction network. Aside from the interaction with Sec61α discussed before, TRAPα interacted strongly with its complex partner TRAPβ (91 RLU). Biochemical purifications identified the TRAP complex as a hetero-tetramer with the four subunits α-δ in a 1:1:1:1 stoichiometry [40,69,70]. Interestingly, using an appropriate pair of constructs the BiLC approach can also identify homo-oligomers, which we found for TRAPα, but not TRAPβ or Sec61α (Figure 8A–C). Based on the bimolecular design of the assay, it is unclear if two TRAPα molecules from adjacent TRAP complexes interact or if TRAPα forms a dimer or even a multimer within the same TRAP complex. Consequently, it will be intriguing to check if the TRAPα/β volume in the ER lumen defined by in situ structures of the mammalian TRAP complex could accommodate for more than one TRAPα protein [14,15,62].

In sum, luminescence complementation provided multiple interactions of subunits of the ER protein translocase in living cells (Figure 8D). Ease of use and expandability of the set of constructs plus the combination of the system with biological and synthetic agents will help to examine the dynamics of ER membrane proteins under different cellular conditions or in different cell lines (Appendix A).

## 4. Materials and Methods

### 4.1. Creation of a Plasmid Library

To clone the different reporter constructs four different backbone plasmids (X-N_S_, X-N_L_, X-C_S_, X-C_L_) for the insertion of cDNA were used. Backbone plasmids as well as the three control plasmids encoding for rPKA-C_L_, cPKa-C_S_, and Halo-C_S_ were provided as part of the NanoBiT PPI Starter System (Promega, Madison, WI, USA). To insert cDNA encoding for a protein of interest a standardized workflow was used. Forward and reverse primers were designed that anneal to the 5′- and 3′-end of the cDNA (Appendix A). Primer overhangs included restriction sites for subcloning of cDNA into the multiple cloning site of the backbone vectors. After purification of the PCR product with QIAquick PCR purification kit (Qiagen, Hilden, Germany) the cleaned PCR product and backbone plasmid were double digested for directed insertion via sticky ends. Fragments were separated on agarose gels and purified via the QIAquick gel extraction kit (Qiagen). Insert and plasmid were ligated in a 3:1 ratio for 1 h at room temperature using T4 ligase (Thermo Fisher, Waltham, MA, USA). The ligation product was used for heat-shock transformation (42 °C for 90 s) of JM101 *E. coli* cells, which were plated on a 100 µg/mL ampicillin/LB-agar plate. After 16 h single colonies were picked and grown in a 2 mL liquid culture using 100 µg/mL ampicillin/TB medium. Cultures were harvested by centrifugation in a table-top centrifuge (3000 rpm, 5 min) and plasmid DNA was purified using the PureYield Plasmid Miniprep System (Promega). All plasmids were verified by sequencing (LGC Genomics, Berlin, Germany) and subsequently amplified in the DH5α *E. coli* strain after heat shock transformation (42 °C for 45 s). Plasmids from a 100 mL ampicillin/LB medium culture were purified using the Plasmid Midi Kit (Qiagen).

### 4.2. Cell Culture and Western Blot

Cell culture experiments were performed using HeLa ATCC no. CCL2 cells. The cells were cultivated in standard DMEM + GlutaMAX media with 10% FCS and 1% Pen/Strep (Thermo Fisher) in a humid environment at 37 °C and 5% CO_2_. To prepare samples for Western blot 6 × 10^5^ cells were seeded in a 6 cm dish (GreinerBioOne, Frickenhausen, Germany) and cultivated for 24 h before plasmid transfection. Amounts of 190 µL Opti-MEM (Thermo Fisher), 2 µL DNA (1 µg/µL), and 8 µL FuGeneHD (Promega) were mixed and incubated for 10 min at room temperature. The transfection mixture was added dropwise to cells, which were refreshed by 4 mL media beforehand. After 24 h cells were washed with PBS (Thermo Fisher), trypsinized (Trypsin-EDTA, Thermo Fisher) and harvested in 2 mL of KHM buffer (110 mM potassium acetate, 2 mM magnesium acetate, 20 mM HEPES/KOH, pH 7.2) plus 125 µg/mL trypsin inhibitor (MP Biomedicals, Illkirch-Graffenstaden, France). Cells were counted with the Countess Automated Cell Counter (Invitrogen, Darmstadt, Germany) and semi-permeabilized as described before [71]. Samples were mixed with Laemmli buffer (60 mM Tris/HCl pH 6.8, 10% (*v/v*) glycerol, 2% (*w/v*) SDS, 5% (*v/v*) 2-mercaptoethanol, 0.01% (*w/v*) bromophenol blue), and denatured at 56 °C for 10 min. A total of 2 × 10^5^ cells were loaded on an SDS-PAGE gel, followed by transfer on a PVDF membrane (Merck Millipore, Billerica, MA, USA). Membranes were blocked for 30 min with 3% (*w/v*) BSA (Roth, Karlsruhe, Germany) dissolved in TBS (150 mM NaCl, 10 mM Tris/HCl pH 7.4). Afterwards, the membrane was incubated for 90 min each with primary and secondary antibodies diluted in blocking solution. After every antibody incubation the membrane was washed four times for 5 min using TBS as first and last washing step and TBS-T (TBS plus 0.005% Tween-20 (Sigma-Aldrich, Steinheim, Germany)) in between. The membrane was dried and scanned by the Typhoon Trio (GE Healthcare, Uppsala, Sweden). The following peptide was used to raise the primary antibody against the SmBiT tag (CVTGYRLFEEIL). The β-actin antibody (Sigma-Aldrich) was used as loading control. Visualization of primary antibodies was done using ECL Plex goat anti-rabbit IgG-Cy5 (VWR, Radnor, PA, USA) or ECL Plex goat anti-mouse IgG-Cy3 (GE Healthcare) and the Typhoon-Trio imaging system in combination with the ImageQuant TL software version 7.0 (GE Healthcare, Uppsala, Sweden).

### 4.3. NanoBiT Assay

Per cavity of a white, flat-bottom 96-well plate (GreinerBioOne) 2 × 10^4^ cells were seeded in 100 µL media and cultivated for 24 h in the incubator. Transfection with the reporter constructs was performed without previous media exchange according to Table 1. The transfection mix was incubated at room temperature for 10 min and added to the wells. Per well per plasmid 50 ng of the DNA construct was used. After incubation for 24 h and 5 min before start of the measurement, the medium was replaced with prewarmed Opti-MEM without phenol red (Thermo Fisher) and the 96-well plate was placed in the preheated microplate reader (Tecan Infinite M200). Luminescence was recorded with the following settings: interval 1 min; shaking before every interval; integration time 1000 ms; settle time 50 ms. To activate the luciferase, 20 µL of the 1:20 diluted Nano-Glo Live Cell Assay System (Promega) were added according to manufacturer’s protocol. Substances were added at the indicated timepoints in the figures with the following final concentrations: Digitonin (Merck, Darmstadt, Germany), 0.002%; Forskolin (Sigma-Aldrich), 15 µM; Peptides (see below), 10 pM–100 µM; RNase A (Roche, Basel, Switzerland), 80 µg/mL; Trypsin (Roche), 50 µg/mL. Stocks of digitonin, forskolin, and peptides were dissolved in DMSO (Sigma-Aldrich) whereas RNase A and trypsin were dissolved in H_2_O. Stocks and solvents as negative control were prediluted in Opti-MEM without phenol red.

### 4.4. Peptide Synthesis

Peptides were synthesized with an automated ResPepSL synthesizer (Intavis, Cologne, Germany) using amide Rink resin as the solid phase and Fmoc-protected amino acids (Carbolution, St. Ingbert, Germany) for coupling according to the method of Merrifield [72]. The resin-coupled peptides were cleaved and deprotected with trifluoroacetic acid (Sigma-Aldrich) followed by precipitation with tert-butyl methyl ether (Thermo Fisher) and further analysis and purification by preparative HPLC (Merck, Darmstadt, Germany). Lyophilized peptides (Lyovac GT2, Finn-Aqua, Tuusula, Finland) were stored at 4 °C. Stock solutions of the peptides were prepared fresh and used the same day.

### 4.5. Antibody Generation

Synthesized peptides were coupled to maleimide-activated keyhole limpet haemocyanin via an N- or C-terminal cysteine for 16 h at room temperature. After size exclusion chromatography, the protein fraction was dialyzed excessively with physiological buffer solution (137 mM NaCl, 2.7 mM KCl,10 mM Na_2_HPO_4_, 1.8 MM KH_2_PO_4_, pH 7.4). Rabbits were immunized subcutaneously in 250 µL doses (100 µg antigen/Freud’s Adjuvant) at 14-day intervals. Blood was collected at 8-day intervals after each immunization. The serum was prepared after agglutination by several 10-min centrifugation steps at 4 °C in a table-top centrifuge.

### 4.6. Statistics and Graphical Representation

Graphs were visualized using Sigma Plot 14.0 (Systat Software GmbH, Erkrath, Germany) and Corel Draw Graphics Suite 2018 software (Coral Cooperation, Ottawa, Canada). Statistical comparison of multiple groups was performed using ANOVA in combination with Dunnett’s multiple comparison test whereas two groups were compared based on an unpaired, two-tailed *t*-test. *p*-values are indicated by asterisks with *p* < 0.001 (***) < 0.01 (**) < 0.05 (*) or as non-significant (ns) if *p* ≥ 0.05. For better differentiation, the *p*-value symbols emanating from a t-test are written in italic in bar graphs. The number of repeats for each experiment are indicated by the individual data points shown in the bar graphs.

## Figures and Tables

**Figure 1 ijms-22-10358-f001:**
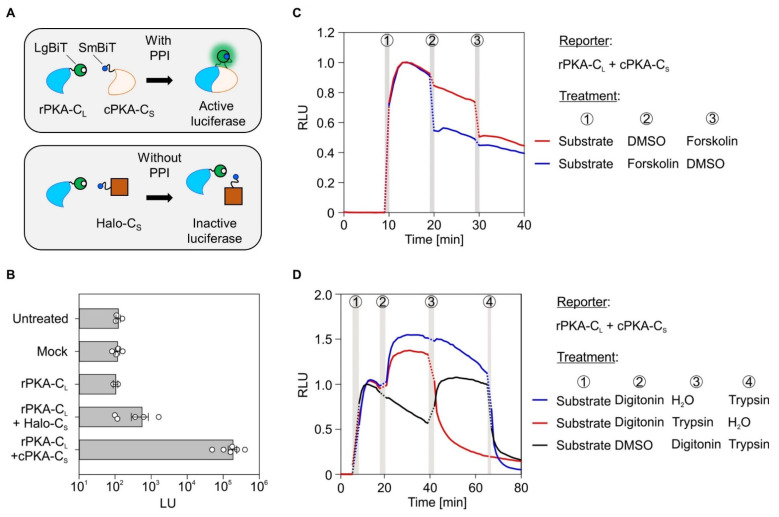
Establishing a bimolecular luminescence complementation assay based on protein kinase A and cell permeabilization. (**A**) Principle of the bimolecular luminescence complementation (BiLC) assay. The upper cell shows the expression of two interacting proteins, the regulatory and catalytic subunit of protein kinase A (rPKA and cPKA), C-terminally (**C**) tagged with the LgBiT (_L_) and SmBiT (_S_) fragment of the luciferase, respectively. Protein–protein interaction reassembles a functional luciferase (glowing green/blue sphere) that emits light upon addition of the substrate furimazine (not shown). In contrast, expression of non-interacting proteins (lower cell) prevents functional reassembly of the luciferase and luminescence. (**B**) In addition to untreated and mock treated cells, total luminescence (LU) was also measured from single or double transfected cells using the rPKA-C_L_, cPKA-C_S_, or Halo-C_S_ fusion constructs. For each condition and the corresponding biological replicates, the signals recorded 4 min after addition of furimazine (this time point usually depicts the peak intensity after furimazine addition, cf. Figure 1C) were plotted. (**C**) Twenty-four hours post transfection with the rPKA-C_L_ and cPKA-C_S_ reporter pair, cells were subjected to the indicated treatment regimens (circled numbers). Nine minutes after starting the measurement, the addition of furimazine stimulated luminescence. After 19 min, cells were treated with forskolin (15 µM) to activate adenylate cyclases and cAMP production, causing disassembly of the PKA subunits. DMSO served as vehicle control. After 29 min, cells were treated in a reciprocal fashion; the ones that received DMSO first now received forskolin and vice versa. Measurements were normalized to the signal intensity recorded 4 min after furimazine application and plotted as relative luminescence units (RLU). (**D**) As in (**C**), but after substrate application, cells were treated with digitonin (0.002%) to permeabilize the plasma membrane and trypsin (50 µg/mL) to digest the reporter and other cytosolic proteins. DMSO and water treatments served as vehicle controls for digitonin and trypsin, respectively. Vertical gray bars in the line diagrams represent manual 1 min application periods without luminescence readings. The dotted lines are extrapolated based on the last and first data points before and after application. C_L_, C-terminally located LgBiT tag; C_S_, C-terminally located SmBiT tag.

**Figure 2 ijms-22-10358-f002:**
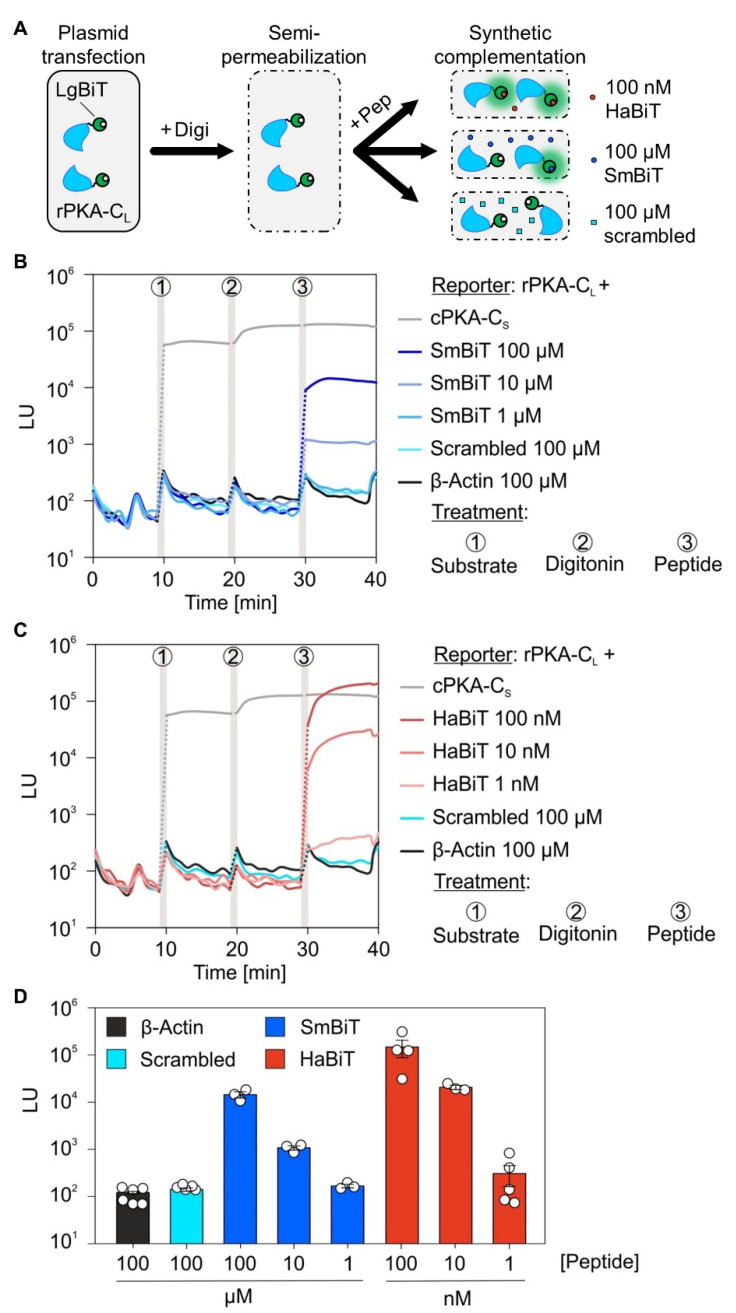
Expression profiling via synthetic complementation using a low (SmBiT) and high affinity (HaBiT) peptide. (**A**) Cartoon of the synthetic complementation approach. Twenty-four hours after transfection of cells with the LgBiT fusion construct (here rPKA-C_L_), the plasma membrane was permeabilized using digitonin and synthetic complementation was achieved by adding one of the chemically synthesized peptides (Appendix AA). The scrambled (100 µM) peptide served as negative control. SmBiT (100 µM) and HaBiT (100 nM) peptides were used as low and high affinity positive controls able to reconstitute the functional luciferase. (**B**,**C**) Twenty-four hours post transfection with rPKA-C_L_, the measurements of luminescence units (LU) were started. At the indicated times, the luciferase substrate was added (treatment 1) and followed by 0.002% digitonin (treatment 2) and the addition of the peptide with the indicated final concentration (treatment 3). The scrambled and unrelated actin peptide were used as negative controls. Cells transfected with the genetically encoded rPKA-C_L_ plus cPKA-C_S_ reporter pair were used for comparison of signal intensity and treated with DMSO instead of peptide as last treatment. Vertical gray bars in the line diagrams represent manual 1 min application periods without luminescence readings and dotted lines are extrapolated based on the last and first data points before and after indicated treatments. Each trace is the average of at least three replicates. Note that traces for the reporter pair as well as the scrambled and actin peptide are the same in (**B**) and (**C**). (**D**) Summary of the recorded signals 4 min after peptide supplementation for each condition and repeat. C_L_, C-terminally located LgBiT tag; Digi, digitonin; Pep, peptide.

**Figure 3 ijms-22-10358-f003:**
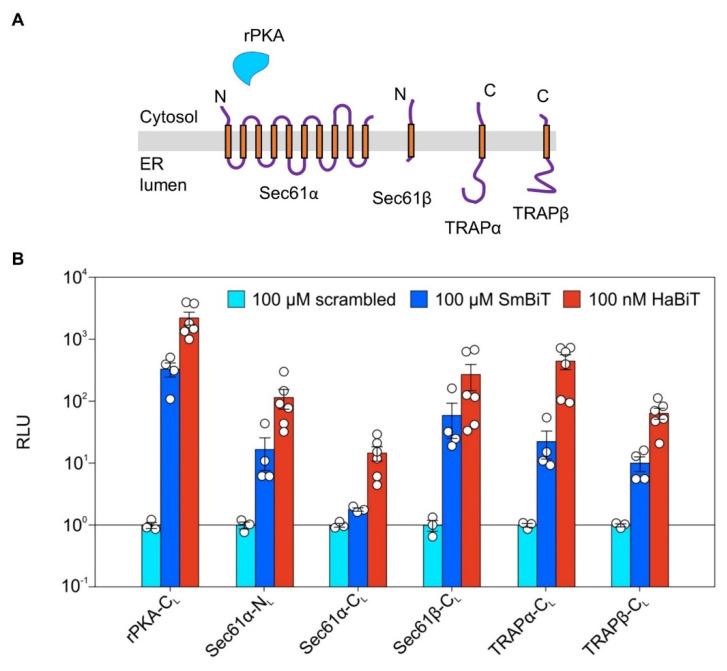
Expression profiles of LgBit-tagged components of the ER protein translocase. (**A**) Topology of the four ER membrane proteins (Sec61α/β, TRAPα/β) used for generating five LgBiT fusions. (**B**) Twenty-four hours post transfection, the expression of the indicated LgBiT fusion constructs tagged at the N- or C-terminus (N_L_ or C_L_) was verified using the synthetic complementation approach with the scrambled, SmBiT, and HaBiT peptide, as demonstrated in Figure 2. The light intensity 4 min after peptide addition was plotted as relative luminescent units (RLU). For each experiment and tested fusion protein, the luminescence measured 4 min after complementation with the scrambled peptide was used for normalization and set to 1. To depict the variation of the individual scrambled peptide readings the average luminescence of this condition was used for its normalization. C_L_, C-terminally located LgBiT tag; N_L_, N-terminally located LgBiT tag.

**Figure 4 ijms-22-10358-f004:**
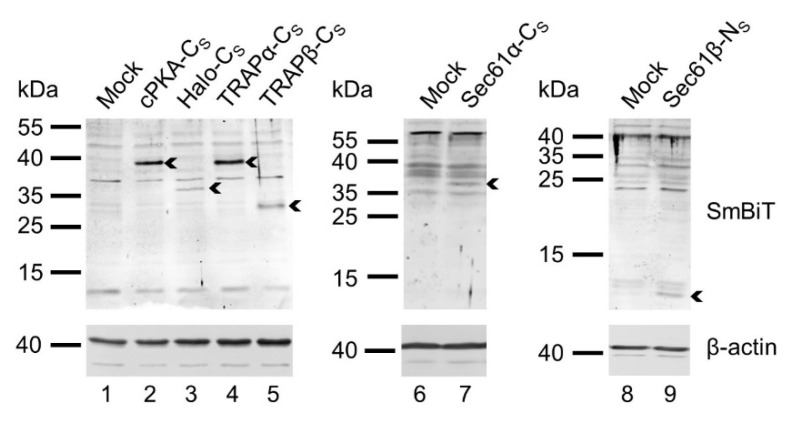
Protein abundance levels of SmBit-tagged components using a tag-specific antibody. Representative Western blot panels for different SmBiT fusion proteins twenty-four hours after transfection of cells are shown. As indicated, lower panels were incubated with anti-β-actin antibody as loading control and upper panels with the in-house generated anti-SmBiT antibody. Despite multiple cross-reactions of the anti-SmBiT antibody (see mock transfected cells in lanes 1, 6, and 8), signals for SmBiT-tagged fusion proteins at the expected molecular weight can be identified (arrowheads). Expected molecular weights: cPKA-C_S_ (42 kDa), Halo-C_S_ (36 kDa), TRAPα-C_S_ (38 kDa), TRAPβ-C_S_ (28 kDa), Sec61α-C_S_ (40 kDa), Sec61β-N_S_ (13 kDa). C_S_, C-terminally located SmBiT tag; N_S_, N-terminally located SmBiT tag.

**Figure 5 ijms-22-10358-f005:**
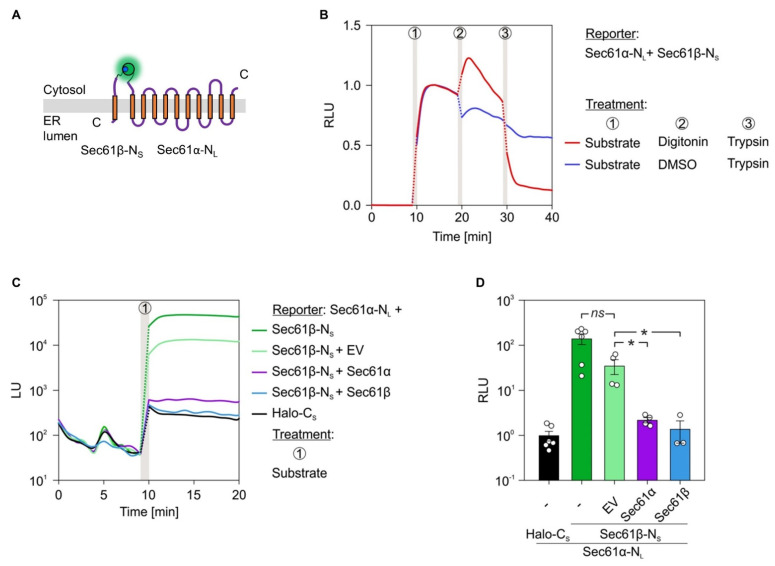
Protein–protein interactions within subunits of the Sec61 complex. (**A**) Topology cartoon of the presumed interaction for the Sec61α-N_L_ and Sec61β-N_S_ pair. As before, PPI reassembles the active luciferase (glowing green/blue sphere) that emits light upon addition of the substrate furimazine (not shown). (**B**) Twenty-four hours post transfection with the Sec61α-N_L_ and Sec61β-N_S_ reporter pair, cells were subjected to the indicated treatments (circled numbers). The furimazine treatment to activate luminescence (after 9 min) was followed by permeabilization using digitonin (0.002%, after 19 min) and trypsin mediated reporter digestion (50 µg/mL, after 29 min). DMSO served as vehicle control for digitonin. Measurements were normalized to the signal intensity recorded 4 min after furimazine application. (**C**) BiLC was combined with competition. Twenty-four hours after double or triple transfection with the indicated constructs, total luminescent units (LU) were measured. Protein names without a suffix do not carry a tag and were designed to compete with the interaction of the tagged interaction partners. (**D**) Measurements from (**C**) were used for quantification. For each condition and the corresponding biological replicates, the signals 4 min after furimazine addition were evaluated and plotted as relative luminescent units (RLU). To do so, the luminescence of the negative control pair (Sec61α-N_L_ plus Halo-C_S_) that was run in parallel for each experiment was used for normalization and set to 1. To depict the variation of negative control pair readings the average LU of this condition was used for its normalization. Statistical comparison of two conditions was based on a student’s *t*-test (indicated by italic labels) and comparison of multiple conditions was done using ANOVA. Vertical gray bars in the line diagrams represent manual 1 min application periods without luminescence readings. The dotted lines are extrapolated based on the last and first data points before and after application. C_S_, C-terminally located SmBiT tag; EV, empty vector; N_L_, N-terminally located LgBiT tag; ns, not significant; N_S_, N-terminally located SmBiT tag. *p*-values are indicated by asterisks with *p* < 0.05 (*) or as non-significant (ns) if *p* ≥ 0.05.

**Figure 6 ijms-22-10358-f006:**
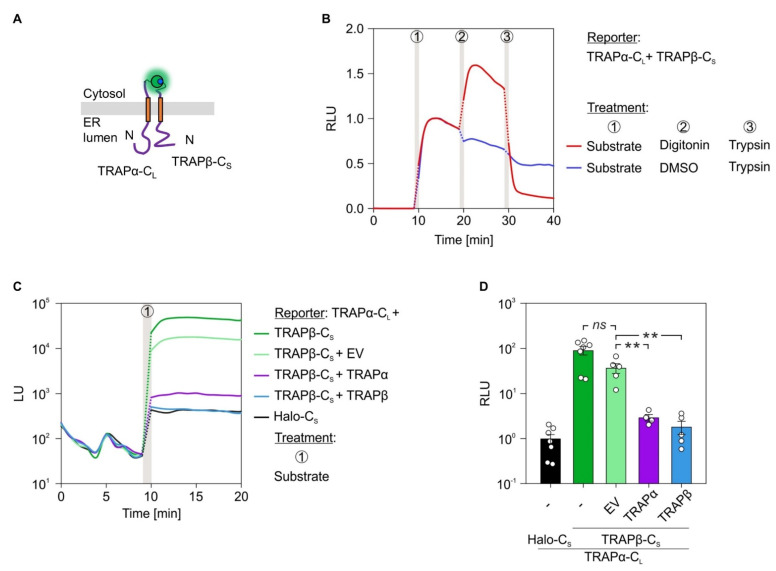
Protein–protein interactions within subunits of the TRAP complex. (**A**) Topology cartoon of the presumed interaction for the TRAPα-C_L_ and TRAPβ-C_S_ pair. Active luciferase is symbolized as glowing green/blue sphere. (**B**) Twenty-four hours post transfection with the TRAPα-C_L_ and TRAPβ-C_S_ reporter pair, cells were subjected to the indicated treatments (circled numbers). The furimazine treatment to activate luminescence (after 9 min) was followed by permeabilization using digitonin (0.002%, after 19 min) and trypsin mediated reporter digestion (50 µg/mL, after 29 min). DMSO served as vehicle control for digitonin. Measurements were normalized to the signal intensity recorded 4 min after furimazine application. (**C**) BiLC was combined with competition. Twenty-four hours after double or triple transfection with the indicated constructs, total luminescent units (LU) were measured. Consistency of the expression efficiency for double and triple transfections was confirmed by synthetic peptide complementation (Appendix A). Protein names without a suffix do not carry a tag and were designed to compete with the interaction of the tagged interaction partners. (**D**) Measurements from (**C**) were used for quantification. For each condition and the corresponding biological replicates, the signals 4 min after furimazine addition were evaluated and plotted as relative luminescent units (RLU). To do so, luminescence of the negative control pair (TRAPα-C_L_ plus Halo-C_S_) that was run in parallel for each experiment was used for normalization and set to 1. To depict the variation of negative control pair readings, the average LU of this condition was used for its normalization. Statistical comparison of two conditions was based on a student’s t-test (indicated by italic labels) and comparison of multiple conditions was performed using ANOVA. Vertical gray bars in the line diagrams represent manual 1 min application periods without luminescence readings. The dotted lines are extrapolated based on the last and first data points before and after application. C_L_, C-terminally located LgBiT tag; C_S_, C-terminally located SmBiT tag; EV, empty vector; ns, not significant. *p*-values are indicated by asterisks with *p* < 0.01 (**) or as non-significant (ns) if *p* ≥ 0.05.

**Figure 7 ijms-22-10358-f007:**
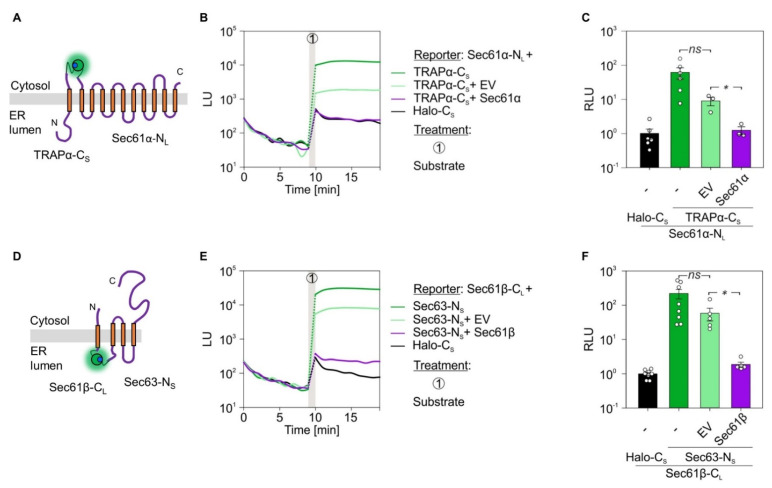
Protein–protein interactions within the holo-translocon: Sec61-TRAP and Sec61-Sec63. (**A**,**D**) Topology cartoon of the presumed interaction pairs Sec61α-N_L_ and TRAPα-C_S_ as well as Sec61β-C_L_ and Sec63-N_S_. Active luciferase is symbolized as glowing sphere. (**B**,**E**) Twenty-four hours post double or triple transfection with the indicated constructs, total luminescent units (LU) were measured. Protein names without a suffix do not carry a tag. Nine minutes into the measurements, furimazine was added to activate luminescence. (**C**,**F**) Quantification and statistical analysis of luminescent readings from (**B**,**E**). For each condition and the corresponding biological replicates, the signals 4 min after furimazine addition were evaluated and plotted as relative luminescent units (RLU). Luminescence of the corresponding negative control pair (Sec61α-N_L_ plus Halo-C_S_ in **C**; Sec61β-C_L_ plus Halo-C_S_ in (**F**)) that was run in parallel for each experiment was used for normalization and set to 1. To depict the variation of negative control pair readings, the average LU of this condition was used for its normalization, respectively. Statistical comparison of two conditions was based on a student’s t-test. Vertical gray bars in the line diagrams represent manual 1 min application periods without luminescence readings. The dotted lines are extrapolated based on the last and first data points before and after application. C_L_, C-terminally located LgBiT tag; C_S_, C-terminally located SmBiT tag; EV, empty vector; N_L_, N-terminally located LgBiT tag; ns, not significant; N_S_, N-terminally located SmBiT tag. *p*-values are indicated by asterisks with *p* < 0.05 (*) or as non-significant (ns) if *p* ≥ 0.05.

**Figure 8 ijms-22-10358-f008:**
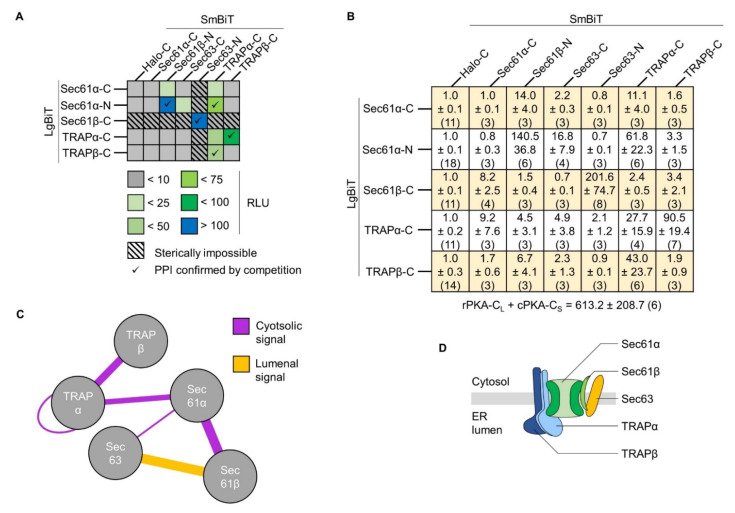
Summary of protein–protein interactions measured in living mammalian cells. (**A**) Heatmap summarizing the tested PPI. Interaction strength is based on relative luminescence units (RLU) normalized to the negative control Halo-C_S_ set as 1 and shown in the first column. RLU below 10 were considered as background (gray color). In addition to the RLU-based interaction strength, the heatmap encodes (i) if a given PPI could be confirmed by competition experiments (checkmark symbols, cf. Figure 5, Figure 6 and Figure 7) and (ii) if the interaction of the tested reporter pair is sterically impossible, i.e., the tags are oriented on different sides of the ER membrane (hachured squares). (**B**) Data summary of tested PPI pairs as average ± standard error of the mean from (n) tested replicates. For better comparison, layout of the grid and heatmap in panel A are identical. (**C**) Interaction network based on the five proteins of interest tested here. While thickness of connecting lines indicates the detected RLU (strength) of the interaction, purple and orange color implies origin of the luminescent signal in the cytosol or ER lumen, respectively. (**D**) Cartoon view of the verified PPI. The suffix -N or -C following protein names indicates N- or C-terminal position of the LgBiT or SmBiT tag.

**Table 1 ijms-22-10358-t001:** Composition and intended purpose of different plasmid transfection mixtures. Stock concentrations of the plasmids were 100 ng/µL. All volumes are provided in microliters.

Transfection Mix	One Plasmid Mix	Two Plasmid Mix	Three Plasmid Mix
Purpose	Synthetic complementation	PPI experiment	PPI competition experiment
Opti-MEM w/o phenol red	7.36	6.72	6.08
LgBit plasmid	0.50	0.50	0.50
SmBiT plasmid	-	0.50	0.50
Competition plasmid	-	-	0.50
FuGENE HD ^1^	0.14	0.28	0.42

^1^ FuGENE HD (Promega).

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
