# Peer review of "Lights, Camera, Interaction: Studying Protein–Protein Interactions of the ER Protein Translocase in Living Cells"

_ijms, 2021, doi:10.3390/ijms221910358_

Round 1

Reviewer 1 Report

The authors used a very well established method (bimolecular luminescent complementation) to demonstrate in live cells protein interactions with various Sec61 components which have been already proposed from structural studies. The approach might be of interest for future studies of protein translocation in ER.

Author Response

We thank reviewer 1 for feedback and comments. The reviewer is absolutely correct with the conclusion that most of the interactions tested and demonstrated here were already visualized by structural studies. We would like to point out that certain interactions, like the one between Sec63 and Sec61α or Sec63 and Sec61β, have not been reported by structural studies for mammalian cells. We think our manuscript lays the cornerstone for future investigations about the dynamics of the mammalian translocon in living cells and hope the reviewer can understand our enthusiasm that such a straightforward, methodically simple, and yet versatile approach will be of interest for future studies of protein translocation.

Reviewer 2 Report

This manuscript validates a cell-based assay that uses a split luciferase reporter to detect protein-protein interactions via biomolecular luminescence complementation (BiLC).  The ability to monitor protein association and dissociation in transfected cells was first established using the regulatory and catalytic subunits of protein kinase A (PKA).  More extensive studies were then conducted using proteins in the Sec61 complex.  These studies also demonstrated that BiLC can be used to establish the topology of interactions between intracellular transmembrane proteins (ie, whether binding involves the cytosolic or lumenal regions of the proteins).  The development and description of this BiLC tool is a valuable addition to the field.  However, as currently written, the manuscript is confusing and in need of improved clarity.

ORGANIZATION

From the first half of the abstract and the first half of the introduction, it appears that the article will be a review of the Sec61 complex.  The remainder of the abstract and introduction are written as if this is a primary research article.  At no point is it clear that proof-of-principle studies for the development of a new assay are the main emphasis of the paper.  The greatest interest in this paper will likely be from investigators who will see potential applications of the BiLC system to their own studies, independent of the Sec61 system.  As such, the authors should re-structure the abstract and introduction to focus on the tool-making goal of the project.  For example, the PKA studies are not mentioned at all in the abstract or introduction.  Essential controls for the assay only receive a cursory mention in the abstract and introduction.  Less information on the Sec61 complex and more information on the problems of monitoring protein-protein interactions in living cells, along with the specifics of assay parameters, will provide a better representation of the work.  A direct mission statement in the abstract might help clarify the goal of the project as well.

The ability to establish the site of binding between two transmembrane proteins (ie, cytosolic vs. lumenal interactions) is a strength of the BiLC system that should be clearly explained in the abstract and introduction.  As documented with the PKA studies, monitoring protein association and dissociation in response to changing cellular conditions is another strength of the system.

Figure 3:  BiLC peptides generated a signal when added to the cytosol of digitonin-permeabilized cells expressing a Sec61beta construct with the complementary BiLC reporter at its C-terminus, which faces the ER lumen.  Digitonin should selectively permeabilize the plasma membrane, leaving the intracellular organelles intact.  As such, the added peptides should not be able to reach their binding partner in the ER lumen.  The description of this Figure in the results reads as if the authors expected a signal for this condition, which was a source of concern when reading the manuscript.  The authors later acknowledge in the discussion that this was not an expected result and provide possible explanations for the discrepancy (p. 14 lines 490 - 497).  This information should be moved to the results section in order to inform the reader that the authors understand the result was unexpected and there are alternative explanations.

Figure 8, which appears to contain new data as well as a summary of the paper's experiments, is only mentioned in the discussion (p. 14 lines 453- 490).  This information should be moved to the results section.  Figure 8 also contains an important control for the BiLC system, showing that two interacting transmembrane proteins with BiLC tags on opposite sides of the membrane (lumenal vs. cytosolic) do not generate a RLU signal.  This information should be provided immediately after describing the results of Figure 7, as Figure 7 only monitors the interaction of transmembrane proteins with BiLC tags on the same side of the membrane.  Thus, it is missing the important negative control presented in Figure 8.

CLARITY

p. 2 lines 81 and 83-84: It is not clear what "topology dependent inter-complex interactions" and "cytosolic PPI could be abolished by semi-permeabilization with trypsin shaving" mean without having read the manuscript. Additional detail should be provided to explain these points. 

p. 3 lines 119-121: "Using the interaction of the PKA subunits…" does not provide sufficient clarity as a stand-alone statement. This sentence only made sense after reading the legend of Figure 1.  Some of the details from the legend should be provided in the text for improved clarity.

The text referring to Figure 1 should mention that trypsin was added for 1 min before continuing with the RLU measurements.  This information is only provided in the figure legend, so I was initially confused as to how trypsin could generate an instantaneous drop in signal when I went directly from the text to viewing Figure 1D.  The procedure for delayed RLU measurement was also a source of confusion when viewing other experiments with protein/peptide addition to digitonin-permeabilized cells (eg, Fig. 2B-C); it should be mentioned in the text for these experiments as well.

When first viewing Figure 1D, there was also a concern that trypsin treatment was reducing the RLU signal of digitonin-permeabilized cells simply by removing cells from the plate.  This concern was addressed later in Figure 5B, but it would be good to at least mention this important control (trypsin treatment of intact cells) when describing Figure 1D in the text.  This would eliminate a possible source of skepticism as individuals read through the manuscript.

p. 3 lines 128-130: It is difficult to infer what the authors mean from the last part of the sentence ("in living cells in combination…"). A more explicit explanation should be given.

p. 12 lines 388-389: The sentence "And two, can the assay…." needs further clarification. There should be an explicit statement that the authors are referring to the interaction of two transmembrane proteins at either the lumenal or cytosolic face of the membrane.

EXPERIMENTAL DESIGN

The authors concluded receptor degradation is responsible for the loss of RLU upon trypsin addition to digitonin-permeabilized cells (Fig. 1D).  This is reasonable, but did the authors add another (non-protease) protein in the place of trypsin to further strengthen their interpretation?

How do the experiments of Figure 2 examine the "stable expression of generated fusion constructs" (p. 5 lines 158-160)?  Stable expression could mean either protein half-life or retention of the transfected plasmid, but Figure 2 focuses on signal strength as a reflection of protein-protein affinity.  Likewise, "stable expression" is used in the description of Figure 3 (p. 8 line 258).  What does this mean?  The cells are transiently transfected, so there is no stable expression.

Legends for Figures 5-7:  In regards to "For each condition and repeat signals", what does repeat signals mean?

Legends for Figures 5-7:  The asterisks in the bar graphs are not specifically mentioned as representing statistical significance, and the p values for significance are not defined.  Does "italic labels" refer to "ns" in the graphs?

Figures 5-7:  Why do the bar graphs (5D, 6D, 7C, 7F) show a range of values for the negative controls when the results for these conditions were, as stated in the legends, set to 1?  Indeed, Figure 8B reports a value of 1.0 +/- 0.0 for these negative controls.

In Figure 7F, the bar graph provides a statistical analysis comparing "-" to "EV" but not "-" to "Sec61beta".

p. 14 lines 458 - 464: the authors should consider transfection efficiency may have an impact on the RLU signal strength - if one of the reporters is poorly expressed, it will not generate a strong signal.

Author Response

Comments and Suggestions for Authors

This manuscript validates a cell-based assay that uses a split luciferase reporter to detect protein-protein interactions via biomolecular luminescence complementation (BiLC).  The ability to monitor protein association and dissociation in transfected cells was first established using the regulatory and catalytic subunits of protein kinase A (PKA).  More extensive studies were then conducted using proteins in the Sec61 complex.  These studies also demonstrated that BiLC can be used to establish the topology of interactions between intracellular transmembrane proteins (ie, whether binding involves the cytosolic or lumenal regions of the proteins).  The development and description of this BiLC tool is a valuable addition to the field.  However, as currently written, the manuscript is confusing and in need of improved clarity.

We very much appreciate the time and effort taken by reviewer #2 to critically assess the data presented and providing such a detailed as well as stimulating feedback. Based on both the concerns raised and the suggestions given for improvement the following point-by-point answers hopefully help to enhance clarity of the manuscript.

ORGANIZATION

From the first half of the abstract and the first half of the introduction, it appears that the article will be a review of the Sec61 complex.  The remainder of the abstract and introduction are written as if this is a primary research article.  At no point is it clear that proof-of-principle studies for the development of a new assay are the main emphasis of the paper.  The greatest interest in this paper will likely be from investigators who will see potential applications of the BiLC system to their own studies, independent of the Sec61 system.  As such, the authors should re-structure the abstract and introduction to focus on the tool-making goal of the project.  For example, the PKA studies are not mentioned at all in the abstract or introduction.  Essential controls for the assay only receive a cursory mention in the abstract and introduction.  Less information on the Sec61 complex and more information on the problems of monitoring protein-protein interactions in living cells, along with the specifics of assay parameters, will provide a better representation of the work.  A direct mission statement in the abstract might help clarify the goal of the project as well.

- We agree with the reviewer and have now emphasized the proof-of-principle aspect as few statements interspersed throughout the manuscript.

- We re-structured abstract and introduction to also focus on the tool-making goal of the project and the challenges of determining bona fide protein-protein-interactions in physiological settings; and did so at the expense of the Sec61 review aspect. As suggested, a mission statement was added to the abstract. However, we kept those changes to a minimum as this manuscript is dedicated to the special issue “Mechanisms of ER Protein Import” and specifically addresses the interactions of ER protein translocase subunits. Where reasonable, we referred the reader to expert reviews on protein-protein-interactions that discuss advantages and disadvantages of existing methods.

- We now mention PKA data in the abstract and introduction and provide more background about the kinase. As the PKA data are not entirely novel and were originally introduced by Dixon et al. 2016 (https://doi.org/10.1021/acschembio.5b00753), a reference we cite on multiple occasions, we did not feel comfortable “advertising” the PKA part more vigorously. While Dixon et al. used another cell type and tested the impact of different temperatures on monitoring the kinetics of the PKA subunit interaction, they also made use of forskolin. We used PKA for introducing the type of BiLC measurements in our cell system (Fig. 1A), verifying the dynamics of PPI influenced by pharmacological modulation (Fig. 1B), comparing negative controls and signal-to-noise ratio of an interaction (Fig. 1C), and introducing the possibility of semi-permeabilization and protease-mediated reporter degradation (Fig. 1D). Overall, we used PKA to get a grasp on the reliability and performance of the BiLC approach.

The ability to establish the site of binding between two transmembrane proteins (ie, cytosolic vs. lumenal interactions) is a strength of the BiLC system that should be clearly explained in the abstract and introduction.  As documented with the PKA studies, monitoring protein association and dissociation in response to changing cellular conditions is another strength of the system.

- As much as we share the opinion with the reviewer about some advantages the BiLC system offers, external readers cautioned us not to overemphasis the option of defining the site of binding or interacting domains of binding partners. One reason is that integral membrane proteins might interact via their transmembrane helices rather than exposed cytosolic or luminal domains. Nevertheless, these soluble domains could come in close proximity providing the chance for reporter complementation and luminescence. Considering the size of tags, linker domains (in our study 12 aa), crosslinkers, or antibodies, many methods like BiLC, FRET, BRET, XL-mass spec, or colocalization via immunofluorescence (confocal or super-resolution) share this ambiguity about spatial juxtaposition versus direct protein-protein-interaction. Yet, to carefully allude at the site of binding of two transmembrane proteins we used terms like “topology-dependent interactions” or “origin of luminescent signal”, which we consider a toned-down version hinting at the same idea as the reviewer suggested. We are in the process of generating variants of membrane proteins with truncations, point mutations, or the SmBiT tag encoded in loops connecting transmembrane helices. With the help of such variants, we might indeed manage to strengthen the argument of the reviewer soon.

- We have re-phrased the term “dynamics of protein-protein-interactions” to “dynamics of protein-protein-interactions in response to changing cellular conditions”

Figure 3:  BiLC peptides generated a signal when added to the cytosol of digitonin-permeabilized cells expressing a Sec61beta construct with the complementary BiLC reporter at its C-terminus, which faces the ER lumen.  Digitonin should selectively permeabilize the plasma membrane, leaving the intracellular organelles intact.  As such, the added peptides should not be able to reach their binding partner in the ER lumen.  The description of this Figure in the results reads as if the authors expected a signal for this condition, which was a source of concern when reading the manuscript.  The authors later acknowledge in the discussion that this was not an expected result and provide possible explanations for the discrepancy (p. 14 lines 490 - 497).  This information should be moved to the results section in order to inform the reader that the authors understand the result was unexpected and there are alternative explanations.

- As suggested, we made the textual rearrangement.

Figure 8, which appears to contain new data as well as a summary of the paper's experiments, is only mentioned in the discussion (p. 14 lines 453- 490).  This information should be moved to the results section.  Figure 8 also contains an important control for the BiLC system, showing that two interacting transmembrane proteins with BiLC tags on opposite sides of the membrane (lumenal vs. cytosolic) do not generate a RLU signal.  This information should be provided immediately after describing the results of Figure 7, as Figure 7 only monitors the interaction of transmembrane proteins with BiLC tags on the same side of the membrane.  Thus, it is missing the important negative control presented in Figure 8.

- As suggested, we moved description of Fig. 8 to the results section and emphasized the importance of the negative controls when reporter pairs are facing different compartments and are physically separated by a lipid bilayer preventing their complementation. The section was placed as new paragraph 2.7 titled “Sterically impossible complementations set the threshold for authentic protein interactions”

CLARITY

  1. 2 lines 81 and 83-84: It is not clear what "topology dependent inter-complex interactions" and "cytosolic PPI could be abolished by semi-permeabilization with trypsin shaving" mean without having read the manuscript. Additional detail should be provided to explain these points. 

- Yes, the reviewer is right. Those statements needed improvement. To enhance clarity, we provided more explanation and the new text passage now reads:

“Our analyses show “intra-complex” interactions between subunits of the Sec61 complex (Sec61α-Sec61β) as well as of the TRAP complex (TRAPα-TRAPβ). Furthermore, we demonstrate “inter-complex” interactions between membrane proteins of different complexes such as Sec61α-TRAPα, Sec61α-Sec63, and Sec61β-Sec63. Both types of PPI, intra- and inter-complex ones, can be perturbed by competitive over-expression of untagged variants. Alternatively, interactions of membrane or soluble protein pairs that rely on reassembly of the split-luciferase components in the cytosol are abolished by the combination of plasma membrane permeabilization and trypsin-mediated reporter degradation.”

  1. 3 lines 119-121: "Using the interaction of the PKA subunits…" does not provide sufficient clarity as a stand-alone statement. This sentence only made sense after reading the legend of Figure 1.  Some of the details from the legend should be provided in the text for improved clarity.

- As requested, some details of the legend from Fig. 1 have been copied to the results section. The new text passage now reads:

“We also tweaked the system for the use of larger, membrane impermeable effectors including soluble enzymes such as trypsin. Cells expressing the rPKA-CL and cPKA-CS reporter pair as representation of a verified PPI were first subjected to permeabilization of the plasma membrane by digitonin and subsequently treated with trypsin for proteolytic cleavage of the reporter proteins to eliminate luminescence.”

The text referring to Figure 1 should mention that trypsin was added for 1 min before continuing with the RLU measurements.  This information is only provided in the figure legend, so I was initially confused as to how trypsin could generate an instantaneous drop in signal when I went directly from the text to viewing Figure 1D.  The procedure for delayed RLU measurement was also a source of confusion when viewing other experiments with protein/peptide addition to digitonin-permeabilized cells (eg, Fig. 2B-C); it should be mentioned in the text for these experiments as well.

- We added two sentences explaining the short measurement “gap” necessary due to the manual application of treatments.

“As depicted by the vertical grey bars in Fig. 1C the addition of furimazine and all other substances required their manual application and caused a short measurement gap that was kept constant using a time window of one minute. The same applies to other measurements shown later.”

When first viewing Figure 1D, there was also a concern that trypsin treatment was reducing the RLU signal of digitonin-permeabilized cells simply by removing cells from the plate.  This concern was addressed later in Figure 5B, but it would be good to at least mention this important control (trypsin treatment of intact cells) when describing Figure 1D in the text.  This would eliminate a possible source of skepticism as individuals read through the manuscript.

- Thank you for pointing this out, changes were made as suggested. We added a short statement about adherent cells and referenced Figs. 5B and 6B for further clarification. The new text passage now reads:

“Of note, trypsin treatment of intact cells prior to permeabilization did not cause a significant drop in luminescence due to reporter degradation or eventual detachment of adherent cells from the microplate surface (cf. Figs. 5B, 6B).”

  1. 3 lines 128-130: It is difficult to infer what the authors mean from the last part of the sentence ("in living cells in combination…"). A more explicit explanation should be given.

- As suggested, the sentence has been re-structured and a more detailed explanation has been added. The new text passage now reads:

“This line of experiments substantiated the applicability and dynamics of the BiLC system for cytosolic interaction partners in living cells. Permeabilization of the plasma membrane by digitonin also granted access of otherwise membrane-impermeable agents including large biological effectors such as trypsin.”

  1. 12 lines 388-389: The sentence "And two, can the assay…." needs further clarification. There should be an explicit statement that the authors are referring to the interaction of two transmembrane proteins at either the lumenal or cytosolic face of the membrane.

- We tried to extend the statement and included the suggested localization of the reporter interaction. Regarding our comment from above (section “Organization”) that reporter association is not inevitably congruent with the true site of interaction of the proteins that they are fused to, we decided to write:

“And two, can the assay detect PPI arising from reconstitution of the split luciferase either in the cytosol or the ER lumen? Detection of luminescence emanating from the ER lumen would further prove the applicability of the system to study PPI between membrane and/or soluble proteins occurring in the ER lumen.”

EXPERIMENTAL DESIGN

The authors concluded receptor degradation is responsible for the loss of RLU upon trypsin addition to digitonin-permeabilized cells (Fig. 1D).  This is reasonable, but did the authors add another (non-protease) protein in the place of trypsin to further strengthen their interpretation?

- Thank you for this suggestion. As we originally outlined in the discussion, we aimed to do so and have now prioritized the proposed experiments for this revision. We added new supplemental information to address this issue. Specifically, we added Fig. S1 that shows that a non-proteolytic enzyme like RNase A does not eliminate luminescence of reporter pairs like rPKA-CL and cPKA-CS (Fig. S1A) or TRAPα-CL and TRAPβ-CS (Fig. S1B).

- In the near future, we are eager to test the effect of apyrase and its impact on the ATP-dependent TAP transporter as channel for the small peptides. Apyrase is an ATP-diphosphohydrolase that catalyzes ATP -> ADP -> AMP + inorganic phosphates. Degradation of ATP should interfere with TAP activity and the transport of small peptides like HaBiT into the ER.    

How do the experiments of Figure 2 examine the "stable expression of generated fusion constructs" (p. 5 lines 158-160)?  Stable expression could mean either protein half-life or retention of the transfected plasmid, but Figure 2 focuses on signal strength as a reflection of protein-protein affinity.  Likewise, "stable expression" is used in the description of Figure 3 (p. 8 line 258).  What does this mean?  The cells are transiently transfected, so there is no stable expression.

- The reviewer is absolutely correct and the statement is, unintentionally, misleading. We exchanged the term “stable expression of generated fusion constructs” to “sufficient expression of fusion constructs and synthesis of the encoded fusion proteins” to avoid evoking the impression of having generated stable, transgenic cell lines. All experiments carried out here were performed using a transient transfection procedure.

Legends for Figures 5-7:  In regards to "For each condition and repeat signals", what does repeat signals mean?

- We apologize if this statement caused confusion. For further clarification we slightly changed the sentences in the figure legends to:

“For each condition and the corresponding biological replicates the signals 4 min after furimazine addition were evaluated and plotted as relative luminescent units (RLU).”

Legends for Figures 5-7:  The asterisks in the bar graphs are not specifically mentioned as representing statistical significance, and the p values for significance are not defined.  Does "italic labels" refer to "ns" in the graphs?

- Instead of providing a definition of the asterisks in each figure legend, we provided a paragraph about “Statistics and graphical representation” in the Material and Methods. There, the description of asterisks and differentiation between italic and non-italicized labels was given.

“Statistical comparison of multiple groups was performed using ANOVA in combination with Dunnett's multiple comparison test and two groups were compared based on an unpaired, two-tailed t-test. P-values are indicated by asterisks with p < 0.001 (***) < 0.01 (**) < 0.05 (*) or as non-significant (ns) if p > 0.05. For better differentiation P-values emanating from a t-test are written in italic in bar graphs. Number of repeats for each experiment are indicated by the individual data points shown in the bar graphs.”

Figures 5-7:  Why do the bar graphs (5D, 6D, 7C, 7F) show a range of values for the negative controls when the results for these conditions were, as stated in the legends, set to 1?  Indeed, Figure 8B reports a value of 1.0 +/- 0.0 for these negative controls.

- We apologize if this statement caused confusion and have extended the indicated figure legends for further clarification. E.g., the legend of Fig. 5D now includes the following statement.

“For each condition and the corresponding biological replicates the signals 4 min after furimazine addition were evaluated and plotted as relative luminescent units (RLU). To do so, the luminescence of the negative control pair (Sec61α-NL plus Halo-CS) that was run in parallel for each experiment was used for normalization and set to 1. To depict the variation of negative control pair readings the average LU of this condition was used for its normalization.”

- Accordingly, we also updated Fig. 8B that now reports the SEM of negative control pairs shown in the bar graphs. Thanks for bringing this to our attention.

In Figure 7F, the bar graph provides a statistical analysis comparing "-" to "EV" but not "-" to "Sec61beta".

- It was surprising to us, that these two conditions were not significantly different from each other despite the almost complete abrogation of the light intensity by bringing untagged Sec61β into the system. This was likely due to the variance of the data points (mind the logarithmically scaled y-axis). We performed one more measurement of this condition and could determine a statistically significant difference which we added for this revision.

  1. 14 lines 458 - 464: the authors should consider transfection efficiency may have an impact on the RLU signal strength - if one of the reporters is poorly expressed, it will not generate a strong signal.

- Again, the reviewer is absolutely correct. We included this expression of concern in the paragraph (now in results section 2.7) and tamed down the statement by starting out with “Undoubtedly, difficult to control methodological constraints such as transfection and expression efficiency of individual cells as well as transfection uniformity across a cell population can cloud deeper interpretation of the luminescence signal intensity of a given PPI. Yet, it is tempting to consider the emitted light intensity as surrogate marker and as first approximation for the strength of a tested PPI.”

- While Figs. 3 and 4 are trying to address the issue of transfection efficiency and expression of reporter constructs, the reviewer`s argument further spurred our curiosity about the issue of transfecting cells in certain cases with multiple plasmids simultaneously. We have added a new Fig. S3 that demonstrates efficient and unaltered expression of fusion constructs even during polytransfections with three plasmids.

We added this observation as new paragraph in the results section 2.5:

“We tested if the expression efficiency of the LgBiT reporter construct TRAPα-CL was impaired by the additional expression of the untagged counterpart TRAPα or the presence of the EV in the transfection mix. The underlying reason was that the addition of a third plasmid during the transfection procedure might reduce transfection and/or expression efficiency of the plasmids. As the exogenously added high-affinity HaBiT peptide was able to substitute for and replace the genetically encoded SmBiT fusion construct of a given reporter pair, the synthetic peptide complementation setup was used to test for expression of the TRAPα-CL construct during transfections with two or three plasmids. As shown in Fig. S3, the presence of an additional plasmid in the mix was not affecting expression of the TRAPα-CL construct which still showed efficient complementation with HaBiT even in presence of the EV or the plasmid encoding the untagged TRAPα protein. Compared to the negative control pair TRAPα-CL and Halo-CS all other three conditions combining TRAPα-CL with TRAPβ-CS alone or in combination with either EV or TRAPα provided the same level of luminescence upon addition of HaBiT (Fig. S3B).”

Granted, this new data set is only testing the impact of transfections with multiple plasmids for the TRAPα-CL construct. Yet, the assay design can be used for testing other plasmid combinations as well in the future.

- Also based on the reviewer`s argument about transfection efficiency and the RLU signal strength as indication for PPI, we compared the performance of three reporter pairs in two different cell lines. As shown in the new Fig. S4, the distribution of RLU units for a non-interacting negative control (TRAPα-CL + Halo-CS), a sterically impossible interaction (TRAPα-CL + Sec63-CS), and a verified PPI (TRAPα-CL + TRAPβ-CS) provide analogous results.

Lastly, we would like to thank the reviewers again for providing critical assessment of the manuscript, the data, and valuable feedback for further improvement!
